# Self-Healing Concrete by Biological Substrate

**DOI:** 10.3390/ma12244099

**Published:** 2019-12-08

**Authors:** How-Ji Chen, Ching-Fang Peng, Chao-Wei Tang, Yi-Tien Chen

**Affiliations:** 1Department of Civil Engineering, National Chung-Hsing University, No. 250, Kuo Kuang Road, Taichung 402, Taiwan; hojichen@dragon.nchu.edu.tw (H.-J.C.); cute1309@gmail.com (C.-F.P.); m19911009@yahoo.com.tw (Y.-T.C.); 2Department of Civil Engineering & Geomatics, Cheng Shiu University, No. 840, Chengching Rd., Niaosong District, Kaohsiung 83347, Taiwan; 3Center for Environmental Toxin and Emerging-Contaminant Research, Cheng Shiu University, No. 840, Chengching Rd., Niaosong District, Kaohsiung 83347, Taiwan; 4Super Micro Mass Research & Technology Center, Cheng Shiu University, No. 840, Chengching Rd., Niaosong District, Kaohsiung 83347, Taiwan

**Keywords:** *Bacillus pasteurii* bacteria, self-healing concrete, crack repair

## Abstract

At present, the commonly used repair materials for concrete cracks mainly include epoxy systems and acrylic resins, which are all environmentally unfriendly materials, and the difference in drying shrinkage and thermal expansion often causes delamination or cracking between the original concrete matrix and the repair material. This study aimed to explore the feasibility of using microbial techniques to repair concrete cracks. The bacteria used were environmentally friendly *Bacillus pasteurii*. In particular, the use of lightweight aggregates as bacterial carriers in concrete can increase the chance of bacterial survival. Once the external environment meets the growth conditions of the bacteria, the vitality of the strain can be restored. Such a system can greatly improve the feasibility and success rate of bacterial mineralization in concrete. The test project included the microscopic testing of concrete crack repair, mainly to understand the crack repair effect of lightweight aggregate concrete with implanted bacterial strains, and an XRD test to confirm that the repair material was produced by the bacteria. The results show that the implanted bacterial strains can undergo Microbiologically Induced Calcium Carbonate Precipitation (MICP) and can effectively fill the cracks caused by external concrete forces by calcium carbonate deposition. According to the results on the crack profile and crack thickness, the calcium carbonate precipitate produced by the action of *Bacillus pasteurii* is formed by the interface between the aggregate and the cement paste, and it spreads over the entire fracture surface and then accumulates to a certain thickness to form a crack repairing effect. The analysis results of the XRD test also clearly confirm that the white crystal formed in the concrete crack is calcium carbonate. From the above test results, it is indeed feasible to use *Bacillus pasteurii* in the self-healing of concrete cracks.

## 1. Introduction

Micro-cracks are an almost inevitable feature of general concrete. Many studies report [1,2,3,4,5,6,7,8,9,10,11,12,13] that some small cracks can be repaired on their own based on the continuous hydration of cement or other physical and mechanical behaviors. This phenomenon is generally called “autogenous healing” or “concrete self-healing” in the concrete world. In the interior of concrete, calcium carbonate precipitation has been considered by most scholars to be the most important factor affecting the autogenous healing of concrete. In nature, the main behavior of the microbiologically induced calcium carbonate crystallization effect is calcium carbonate precipitation.

Many researchers have attempted to incorporate specific healing agents directly into concrete mix to repair cracks by their reaction. However, most of the proposed healing agents are chemical substances that are not environmentally friendly [13]. In recent years, bacterial precipitation caused by bacteria has been considered to be an environmentally friendly and economical material. Therefore, many researchers have proposed the use of bacteria as a self-healing agent and have applied this technique to concrete [14,15,16]. In published studies [1,2,3,4,5,6,7,8], bacterially produced calcium carbonate precipitates have been successfully used to repair concrete or limestone surfaces to enhance concrete strength or durability. However, the bacteria in these studies were applied externally to cracked concrete structures or specimens. Thus, the repair mechanisms in these studies could not be defined as self-healing. Recently, scholars have tried to mix bacterial spores and organic compounds required for their metabolism with fresh cement paste [15,17,18]. The results of these studies also clearly observed the precipitation of calcium carbonate produced by bacteria, but only in early-age concrete specimens (1–7 days). It is presumed that because of the continuous hydration of concrete, the colloidal pores gradually shrink and the bacterial cells or spores are crushed and destroyed during the cement hydration stage [15,17,18]. Therefore, most scholars believe that the addition of bacteria requires the presence of a carrier with a “shell” function to protect bacteria.

The most commonly used strain for biomineralization is *Bacillus pasteurii* DSM33, which is a Gram-positive aerobic bacterium that is ubiquitous in the soil and produces a large amount of intracellular urease. Basically, urease catalyzes the hydrolysis of urea to produce carbon dioxide, ammonia, and hydroxide ions. The reaction equation is as follows:(1)$${\mathrm{NH}}_{2}\left(\mathrm{CO}\right){\mathrm{NH}}_{2}+3H_{2}O\to 2{\mathrm{NH}}_{4}{}^{+}+2{\mathrm{OH}}^{-}+{\mathrm{CO}}_{2}.$$
With the release of hydroxide ions, the pH gradually rises, and carbon dioxide is converted into carbonate ions, which combine with calcium ions in the environment to form calcium carbonate precipitates or gel with other substances [19,20]. The aforementioned reaction is generally referred to as biomineralization. The results of Day et al. [21] showed that in the coexistence of *Bacillus licheniformis*, urea, and calcium chloride, calcium carbonate deposition can be induced to repair concrete cracks. De Muynck et al. [22] pointed out that if bacteria die, but urease activity still exists, the reaction that induces calcium carbonate deposition will continue. Microbiologically Induced Calcium Carbonate Precipitation (MICP) deposition helps to enhance the resistance of water-bearing sandy soil to shear forces [23]. It is also possible to form a protective layer that is more resistant to acid and alkali corrosion on the surface of the stone, and the strength against acid etching can reach a pH of 2.0, further prolonging the life of a historical artifact. Therefore, MICP deposition is considered to be a method for repairing stone monuments [24,25,26]. In addition, Perito and Mastromei [27] pointed out that *Bacillus subtilis* is a *Bacillus* strain that also has the ability to induce calcium carbonate deposition.

The shape of *Bacillus pasteurii* is close to a round rod or a spiral, and its cell diameter is about 0.5 to 3 μm. Studies by Mitchell [28] indicate that *Bacillus* species are more suitable for survival in soil environments with a void size greater than 6 μm. This is also the main reason why bacteria cannot be directly mixed into concrete. Wanga [29] encapsulated *Bacillus sphaericus* spores into a hydrogel and mixed it with normal concrete. The test results were surprising, and the spore viability did not decrease. This proves that specimens containing biohydrogel can fix cracks in concrete. The maximum crack width that the specimen can heal is about 0.5 mm, while the healing crack width of non-biological hydrogels is 0–0.3 mm. The water permeability of the specimen was reduced by about 68%, indicating a significant improvement. Wiktor [17] attached bio-healing agents to porous clay granules and then mixed them with general concrete. The experimental results showed that the concrete mixed with bacteria had a better crack-healing ability. Sierra-Beltran [30] used a strain-hardening cement-based composite (SHCC) with a biological strain implanted inside as a concrete repair mortar. The test results showed that the amount of calcium carbonate precipitate did not significantly exceed that of the control group, and it may be that the biological strains were directly implanted in the mortar and the pore size was too small to continue the survivability of the bacteria.

Certain bacteria (such as *Bacillus*) will form spores with stronger resistance in cells under harsh environments such as drought and freezing. When the environment is suitable, the spores will germinate to produce new bacteria [31,32]. The bacteria that can form spores contain aerobic *Bacillus* and anaerobic *Clostridium* [33]. According to the literature [34], some spores can even sleep for hundreds to thousands of years. Spore staining is used in bacteriology to determine whether bacteria contain spores. The methods commonly used for spore staining are the Schaeffer–Fulton stain and Moeller stain [35,36]. Bacterial species can be easily identified by spore staining. Khoury et al. [37] mentioned that bacteria can kill bacterial vegetative cells at temperatures between 60 and 90 °C, and this temperature range is sufficient to induce sporulation.

In view of the above, this paper aims to take advantage of the biological activities inherent in nature using biomineralization technology to study the crack-healing ability of self-healing concrete by biological substrates. In the study, the bacteria are first sporulated, and in order to protect the strain, the biological species are fixed in a porous lightweight aggregate and then added to the concrete. In this way, the porous lightweight aggregate not only represents the aggregate in the concrete but also serves as a protective container for bacterial spores. Such a system can greatly improve the feasibility and success rate of bacterial mineralization in concrete.

## 2. Experimental Details

### 2.1. Test Program 

In this study, *Bacillus pasteurii* was first treated to a spore state by means of temperature increase. Then, field emission scanning electron microscopy (FE-SEM) observations and spore staining tests were used to verify whether sporulation occurred. After the spore-treated strains were placed in the lightweight aggregate, the spore-forming strains were re-activated to restore their activity for subsequent concrete crack repair testing.

The lightweight aggregates of the control group were not implanted with biological species and were cured in air, while the lightweight aggregates of the experimental group were implanted with biological species. Among them, the air-cured concrete specimens were regarded as experimental group I, and the cycle-cured concrete specimens were regarded as experimental group II. The crack healing ability of the cracked concrete specimens was examined by a crack microscopic test and an X-ray powder diffraction test. 

### 2.2. Materials

The materials used in this study were as follows:Cement: a locally produced Type I Portland cement with a specific gravity of 3.15 and a fineness of 3400 cm^2^/g;Water: general tap water;Fine aggregate: natural sand with a particle size of less than 1 mm;Lightweight aggregate: The lightweight aggregate used was made by crushing natural shale and then firing at 1100–1200 °C. Its surface was porous and irregular in shape with a particle size of 1–8 mm, as shown in Figure 1. The basic properties of the artificial lightweight aggregate are shown in Table 1;Reinforcement: #4 rebar was applied to the beam specimen;*Bacillus pasteurii* (DSM 33): the strain number was BCRC11596, and it was ordered from the Taiwan Food Industry Development Institute;Calcium lactate: this was implanted in lightweight aggregates as a nutrient source for *Bacillus pasteurii*;Yeast extract (YE): this is in powder form and contains a wide variety of vitamins, minerals, amino acids, etc., which are widely used as nutrients, and its leachate can be used as a medium with considerable efficacy;Calcium acetate: this was used as a source of calcium ions during curing;Urea: this was used as source of carbonate ions during curing.

### 2.3. Strain Implantation and Mix Proportions

The steps used for implanting the strain into lightweight aggregates were as follows:(1)Immerse the treated lightweight aggregates into a nutrient source (calcium lactate 80 g/L, yeast extract 1 g/L) for 30 min.(2)After the soaking is completed, drain and take out the lightweight aggregates and place them in an oven at 37 °C for 5 days.(3)Repeat steps (1) and (2) once, and immerse the lightweight aggregates in the nutrient source twice.(4)Dip the lightweight aggregates containing the nutrient source into the bacterial spore solution for 30 min.(5)Drain the soaked lightweight aggregates and place them in an oven at 37 °C for 5 days to complete the work of strain implantation in lightweight aggregates.

After the strains were implanted into the lightweight aggregate (as shown in Figure 2), concrete mixing was carried out. The water-to-cement ratio of the mixed concrete was 0.60, and the detailed proportioning design is shown in Table 2. After the concrete was mixed, 48 cylindrical specimens (100 mm in diameter × 200 mm in height) and 48 beam specimens (3600 mm in length × 100 mm in width × 100 mm in thickness) were cast and compacted using an external vibrator. After casting, all the specimens were covered overnight with wet hessian and polyethylene sheets for a period of 24 hours. After 24 hours, the specimens were demolded. 

The specimens of the control group were lightweight aggregate concrete without implanted bacteria or a nutrient source. The curing method involved placing the test specimens directly in the curing room for air curing, as shown in Figure 3a. The temperature of the curing room was maintained at 23 ± 1.5 °C, and the humidity was 100%. The specimens of experimental group I were implanted with bacteria and a nutrient source. The curing method was the same as that of the control group. The specimens from experimental group II were immersed in a mixed aqueous solution of urea and calcium acetate (see Figure 3b), wherein the mixed aqueous solution contained urea at a concentration of 1 mole and calcium acetate at a concentration of 0.5 moles. The specimens were alternated between curing in a mixed aqueous solution and air. That is to say, the specimens were immersed in an aqueous solution for 1 day, and then the specimens were taken out and placed in the air for 1 day, and the cycle number was 2 days until the required age, and then they were taken out and observed.

All specimens were cured until the day before the test age. After the specimens were taken out of the curing tank, they were kept in a dry state, and various tests were carried out to verify the self-healing effect of the concrete.

### 2.4. Instrumental Setup and Test Procedures

The microscopic state of the strains before and after the heat treatment was observed by field emission scanning electron microscopy (FE-SEM) (JEOL JSM-7401F, Tokyo, Japan) to understand the changes in strain appearance and spore culture. The mode of SEM observation in the experiment was secondary electron imaging, and the image was presented by an upper detector (SEI) and a lower detector (LEI), depending on the type and position of the detector.

In this study, powder samples were randomly taken from inside the concrete specimen and near the surface of the crack. A component analysis of powder samples was performed by a high-resolution X-ray diffractometer (PANalytical X’Pert Pro MRD, Almelo, the Netherlands). When the calcium carbonate crystal was irradiated with X-rays, the angles (Theta) reflected by the X-rays were mainly 29.3°, 39.5°, and 47.3°. When calcium hydroxide was irradiated with X-rays, the angle of reflection by X-ray irradiation was mainly 34.1°. As for Quartz, when X-rays were irradiated, the angles reflected by X-rays were mainly 26.4° and 69.1°.

## 3. Results and Discussion

### 3.1. Results of the Sporulation of the Bacillus Pasteurii Strain

From the results of the photographs shown in Figure 4 and Figure 5 of the unheated *Bacillus pasteurii* in the SEI or LEI shooting mode, the surrounding environment shows that there was no other impurity in the *Bacillus* and that the surface of the *Bacillus* was smooth and there was no adhesion phenomenon. After heating, there were many white spots (SEI) and particles (LEI) in addition to the *Bacillus* itself around the *Bacillus pasteurii* image. Moreover, the appearance of the *Bacillus* became more and more uneven, and in some cases, the *Bacillus* was observed to have protrusions and particle adhesion on the surface.

It can be seen from the image results that the unheated *Bacillus pasteurii* was flat and clean under the conditions of SEI or LEI, and the appearance of the *Bacillus* was also different. After heating, the appearance of the *Bacillus* was broken and there were more impurities in the image. In a high-temperature environment, in order to cope with the severe environment, *Bacillus pasteurii* will produce spores to continue life. It can be speculated that the white spots (SEI), particles, and protrusions (LEI) in the image after heating may be spores, because the spores of *Bacillus* will be released from the broken sporangia when they mature. The image obtained by the FE-SEM observation was the only appearance of a living thing. In order to confirm that the observed image was a spore of *Bacillus pasteurii*, the spore staining method was used for observation. The spore staining method mainly uses malachite green solution to stain the spores, and safranin solution is used to infect the cell wall of the *Bacillus*, and this is then observed by an optical electron microscope at 800 times magnification.

The image results of Figure 6 and Figure 7 show that the unheated *Bacillus pasteurii* was densely distributed under microscope observation, and the *Bacillus* body was red, while the heated *Bacillus pasteurii* was loosely distributed and exhibited a green, non-rod shaped cell. From the experimental results, it was found that the number of cells of the *Bacillus* was less after heating than when the *Bacillus* was not heated. It is speculated that when the bacteria respond to the harsh environment, after depositing their own DNA into the spore, they will die by themselves to continue life. The green spots observed in Figure 7 can be presumed to be spores released by the *Bacillus*.

According to the results of the FE-SEM observations and spore staining, it can be inferred that the test can indeed make the bacteria produce spores.

### 3.2. Results of the Reactivation Confirmation Test of the Bacillus Pasteurii Strain

The reactivation test was carried out to verify whether the spores were germinated into *bacilli* by re-cultivation after the *Bacillus pasteurii* was heated to form spores. The test results are shown in Figure 8. It was observed that the spore culture solution after the culture showed a turbid state, and the optical density (OD) value was found to be about 1.0. The OD values of the freshly cultured bacterial liquid, the spore-formed bacterial liquid, and the bacterial culture cultured after sporulation were 1.2, 0.7, and 1.0, respectively. It is considered that the bacterial liquid after sporulation is active from the change in the OD value, and the turbidity of the medium solution indicates the activity trace of the organism.

After the spore-formed *Bacillus pasteurii* was planted into the aggregate as a biological strain, it was necessary to further understand whether the nutrient source could still be provided to restore the activity again. The test results are shown in Figure 9. It was observed that the medium solution containing the biological strains showed a turbid state, and the OD value was about 1.0, while the OD value of the original culture solution was 0 (control group). According to the test results, the *Bacillus pasteurii* spores can be re-germinated after being stored in the lightweight aggregate.

For the strains revived by the lightweight aggregates containing biological species, the urease production and activity was further confirmed. The test results are shown in Figure 10. It can be observed by the naked eye that there were crystal precipitates at the bottom of the test tube. Therefore, it can be estimated that the urease-containing *Bacillus pasteurii* strain will still be released after activation. Therefore, the MICP reaction should be possible.

The urease test was carried out on strains cultured from lightweight aggregates containing biological species. On the one hand, this was done to determine whether *Bacillus pasteurii*, which becomes spores, can restore its activity again by giving nutrients. On the other hand, it confirmed that the re-cultivation of the bacterial spores in lightweight aggregates restores the activity. From the preliminary biological tests of the lightweight aggregate re-culture test and the urease test containing biological species, it is known that the use of lightweight aggregate as a carrier of the strain is feasible. The strain is still active and forms a specific chemical reagent (urea + calcium acetate), which can successfully induce the urease action of the bacteria to form a calcium carbonate precipitate.

### 3.3. Results of Microscopic Testing of Cracks in Self-Healing Concrete

In the crack repair test of lightweight aggregate concrete, two different specimen types were planned. One was a cylindrical specimen, and the other was a beam specimen (the center is placed with #4 steel bars to prevent breakage, as shown in Figure 11). For these two different types of concrete specimens, the crack had a design width of about 0.1–2.0 mm. After the concrete specimens to be observed were cured in different ways, at the ages of 1, 3, 7, 14, 21, 28, 56, and 91 days, they were taken out for microscopic observation.

#### 3.3.1. Observation of Cracks in Cylindrical Specimens

This study used a splitting method (as shown in Figure 12) to create deep and wide cracks in cylindrical specimens. The width of the produced crack was 0.1–2.0 mm, as shown in Figure 13, Figure 14 and Figure 15. It can be seen from the test results that the control group and experimental group I had no repair phenomenon, regardless of the width of the crack in the sample. In contrast, in experimental group II, with the increase in the curing age, the crack of the sample began to show obvious repair on the 28^th^ day of curing (as shown in Figure 15).

#### 3.3.2. Observation of Flexural Cracks in Beam Specimens

Under a controlled load, the beam specimens produced flexural cracks (as shown in Figure 16). The width of the cracks ranged from 0.1 to 2.0 mm, as shown in Figure 17 and Figure 18. Careful observation revealed that the resulting cracks were wider near the surfaces of the specimens and thinner near the middle of the specimens. As with the test results of the cylindrical specimens, the control group and experimental group I did not show a self-healing phenomenon regardless of the crack width of the specimens. In contrast, in experimental group II, the cracks in the specimen began to show significant self-healing on the third day of curing. As the number of curing days increased, the crack self-healing effect was better, regardless of the crack width.

As can be seen from Figure 16, for a relatively small deflection crack, for example, less than 0.1 mm, the crack can be completely repaired after about the 14th day. In addition, the inclusion of steel in the specimen allowed it to not crack too much when cracks were formed. Also, because there was a steel strip to pull the cracked sample, *Bacillus pasteurii* was less susceptible to disturbance when it produced calcium carbonate precipitate for crystal stack repair. If the flexural crack width reached about 1.2 mm (Figure 17), the calcium carbonate crystal completely filled the crack when the curing age reached 91 days. As with the splitting crack observation of the cylindrical specimen, when the flexural crack was wider, the crack initiation time and the crack repair time were longer.

#### 3.3.3. Fracture Repair Observation of the Specimen Profile

To further understand the effectiveness of biobacterial mineralization and accumulation, this study compared the crack profiles of the specimens. As is clear from Figure 19, the experimental group II specimen-implanted biological strains were covered with calcium carbonate crystals in contrast to the control group specimen. Further, the cross-section of the experimental group II specimen was observed with a portable microscope. As can be seen in Figure 20, calcium carbonate crystals were formed outward from the surface of the aggregate. This result indicates that calcium carbonate crystals were produced by biological species implanted in the aggregate.

According to the test results of the crack self-healing of the concrete cylindrical specimens and beam specimens, it is known that experimental group II of the circulating curing method can use the biological strains to repair the cracks in the concrete. In order to further understand the repairable depth of the crack after the action of the biological strains, the experimental group II concrete specimen was cut along the vertical plane of the crack, and the vertical section was observed. The observed crack width was approximately 0.5 mm (as shown in Figure 21). The vertical section observation results are shown in Figure 22 and Figure 23 after 91 days of crack self-healing. The repaired depth of the crack was about 1.2 mm from the surface of the specimen, as shown in Figure 22. As can be seen from Figure 23, the calcium carbonate crystals produced by the biological species were very fine and compact, and the cracks were repaired perfectly, preventing the concrete from being damaged by external foreign matter invasion. It is worth noting that, as can be seen from the figure, when *Bacillus pasteurii* was repaired with calcium carbonate crystals, a large amount of calcium carbonate precipitate formed from the outer edge of the crack. After sealing the outer edge of the crack, it extended to the inside and continued to slowly deposit calcium carbonate crystals for repair. 

### 3.4. Verification of the Product: Calcium Carbonate Crystal

It can be seen from Figure 24 and Figure 25 that the lightweight aggregate concrete containing biological species had calcium carbonate crystals at the center or near the surface. Because the concrete was mixed with fine sand, the X-ray reflection value of quartz is also shown in the figure. Basically, the intensity of the X-ray reflection can indicate the amount of crystal present. It can be seen from the figure that the experimental group I samples of air curing or the experimental group II samples of circulating curing had peaks of calcium carbonate crystals on the surface that were larger than the center. Therefore, it is foreseeable that the calcium carbonate crystal content measured at the center of the sample was not high. The reason for this may be that under long-term curing, the cement hydration reaction tended to be complete so that the interior of the concrete became denser. As a result, the MICP reaction of the strain was limited, and the amount of calcium carbonate was less. 

Moreover, in the experimental group II samples, *Bacillus pasteurii* also provided additional calcium carbonate formation on the crack surface. The concrete specimen did nothing more than form a protective layer, which made it difficult for oxygen to enter the sample, thus causing the internal *Bacillus pasteurii* activity to be reduced, and it was difficult to form calcium carbonate crystals. Therefore, the calcium carbonate crystal formed at the center of the sample was smaller than the crystal on the surface of the sample. It can also be seen from the figure that the peak of the calcium carbonate crystal of the experimental group II samples at both the center and the surface was higher than that of the experimental group I samples. This result is consistent with the previous observations, which means that experimental group II specimens can produce significant calcium carbonate precipitation for crack repair.

## 4. Conclusions

According to the test results and analysis, the comprehensive conclusions are as follows:According to the results of the FE-SEM observations and spore staining, it can be inferred that *Bacillus pasteurii* can be made to sporulate by heating.From the *Bacillus pasteurii* spore culture medium and the re-cultivation of lightweight aggregates containing biological bacteria, it can be seen that the *Bacillus pasteurii* spores in the lightweight aggregate can be reactivated into bacteria by re-administering nutrients to restore activity, and the urease test confirmed that bacteria in the lightweight aggregate could induce MICP to produce calcium carbonate precipitate.From the crack repair test, it is known that the lightweight aggregate concrete containing biological bacteria requires additional urea and calcium source maintenance to repair the 0.1–2.0 mm wide crack. Concrete nutrients cured in the air are not sufficient to provide *Bacillus pasteurii*, so the repairability of concrete cracks is not as expected.According to the observations of the crack profile and the depth of crack repair, it is known that the calcium carbonate precipitate produced by the action of *Bacillus pasteurii* starts from the boundary between the aggregate and the cement paste, fills the entire fracture surface, and then deposits to a certain thickness. Thus, the repair effect of the concrete can be formed.The results of the XRD analysis confirmed that the white crystal formed in the concrete crack was calcium carbonate, and the closer to the surface of the specimen, the higher the calcium carbonate content. Because the bacterial growth environment at the surface is better, the MICP reaction is more obvious. Therefore, when *Bacillus pasteurii* is repaired with calcium carbonate crystals, a large amount of calcium carbonate precipitate is formed from the outer edge of the crack, and the outer edge of the crack is sealed. Further extending to the inside, the calcium carbonate crystals are slowly accumulated for repair.This study demonstrates that the use of lightweight aggregate as a carrier and the implantation of the *Bacillus pasteurii* strain can induce MICP and produce calcium carbonate crystals to fill small cracks in concrete. The *Bacillus pasteurii* used in this study and the calcium carbonate formed by it have been confirmed to be harmless to the human body, so it should be feasible to use this method in the self-healing of concrete cracks.

## Figures and Tables

**Figure 1 materials-12-04099-f001:**
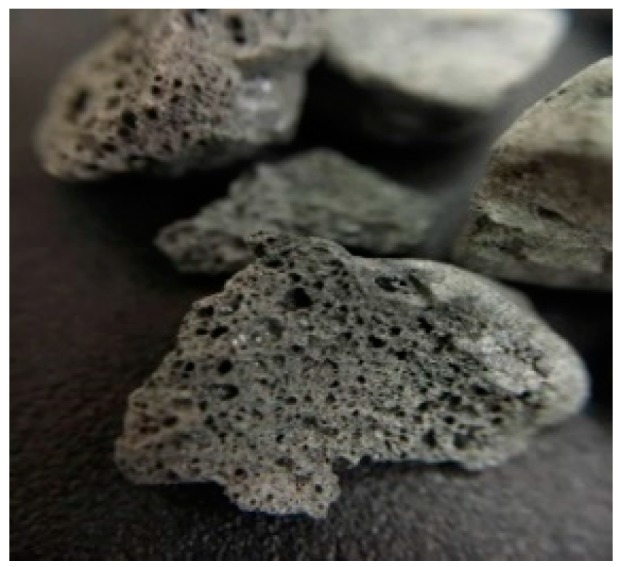
Appearance of lightweight expanded shale aggregates.

**Figure 2 materials-12-04099-f002:**
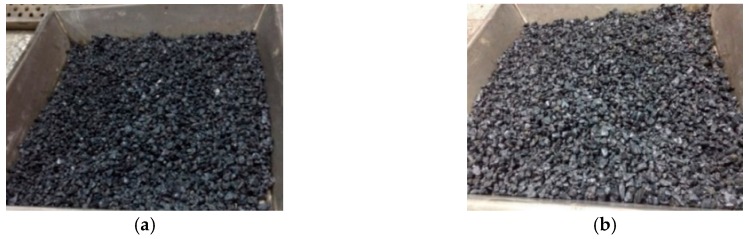
Lightweight aggregate containing biological species: (**a**) soaked once; (**b**) soaked twice.

**Figure 3 materials-12-04099-f003:**
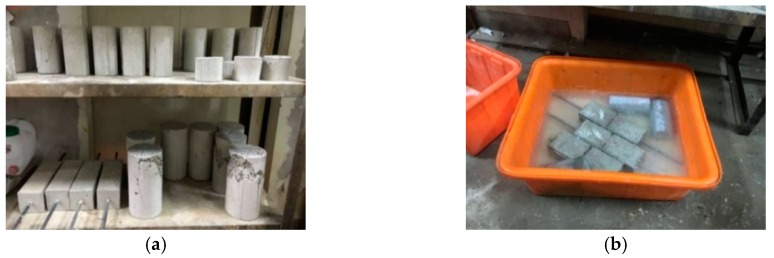
Curing situation of specimens: (**a**) control group and experimental group I; (**b**) experimental group II.

**Figure 4 materials-12-04099-f004:**
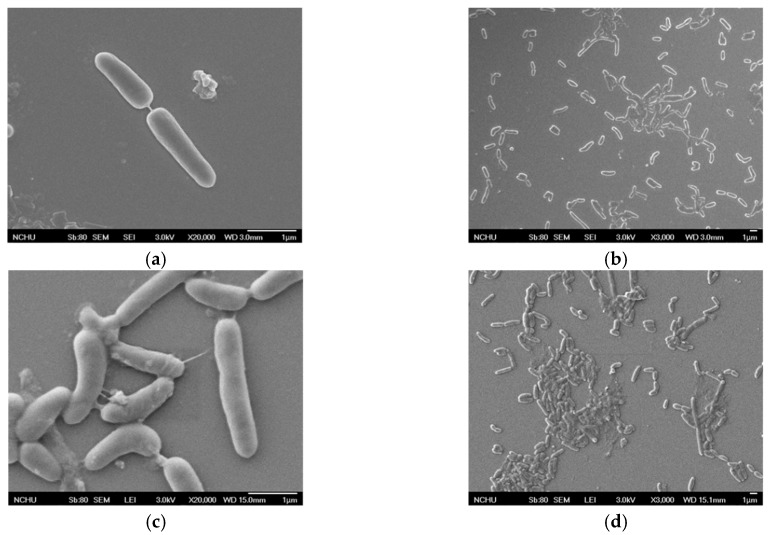
Unheated *Bacillus pasteurii* images: (**a**) SEI (upper detector image, 20,000 times magnification); (**b**) SEI (3000 times magnification); (**c**) LEI (lower detector image, 20,000 times magnification); (**d**) LEI (3000 times magnification).

**Figure 5 materials-12-04099-f005:**
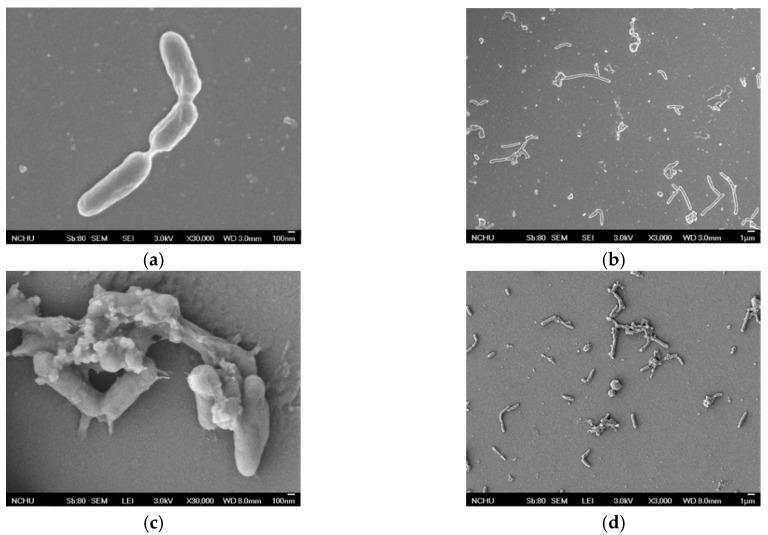
Heated *Bacillus pasteurii* images: (**a**) SEI (20,000 times magnification); (**b**) SEI (3000 times magnification); (**c**) LEI (20,000 times magnification); (**d**) LEI (3000 times magnification).

**Figure 6 materials-12-04099-f006:**
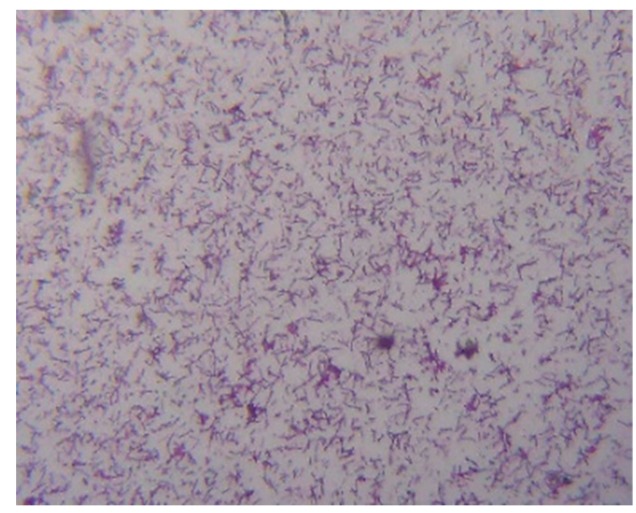
Unheated *Bacillus pasteurii* image (stained).

**Figure 7 materials-12-04099-f007:**
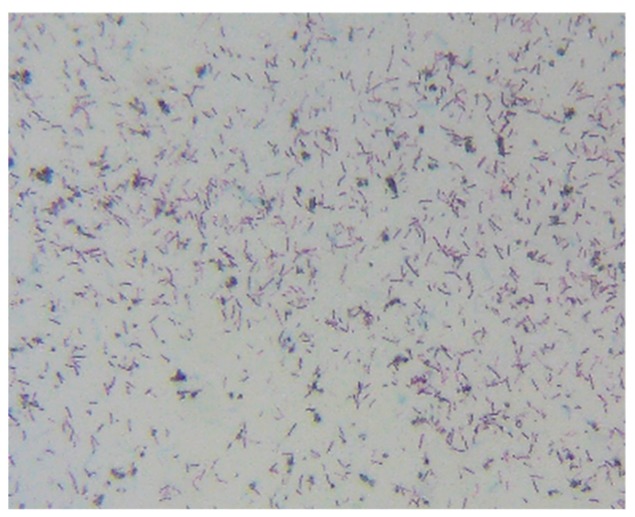
Heated *Bacillus pasteurii* image (stained).

**Figure 8 materials-12-04099-f008:**
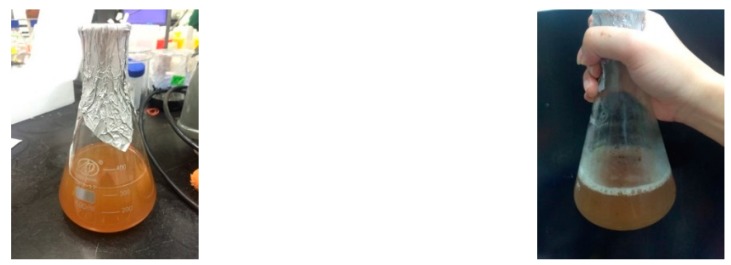
Confirmation of activity after sporulation of the strain (replanting of bacterial solution).

**Figure 9 materials-12-04099-f009:**
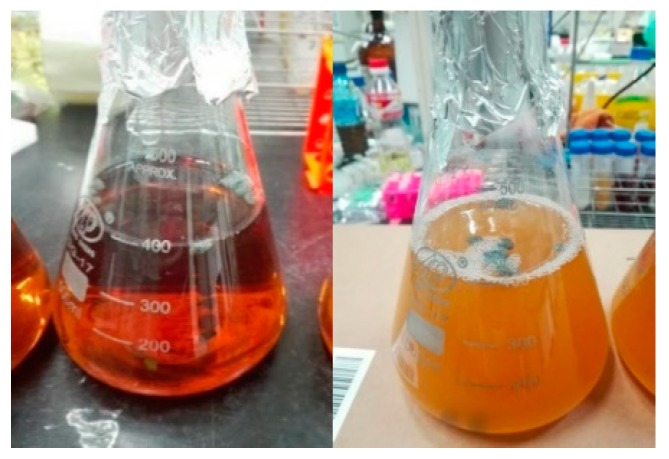
Confirmation of activity after sporulation of the strain (replanting of aggregate containing biological species).

**Figure 10 materials-12-04099-f010:**
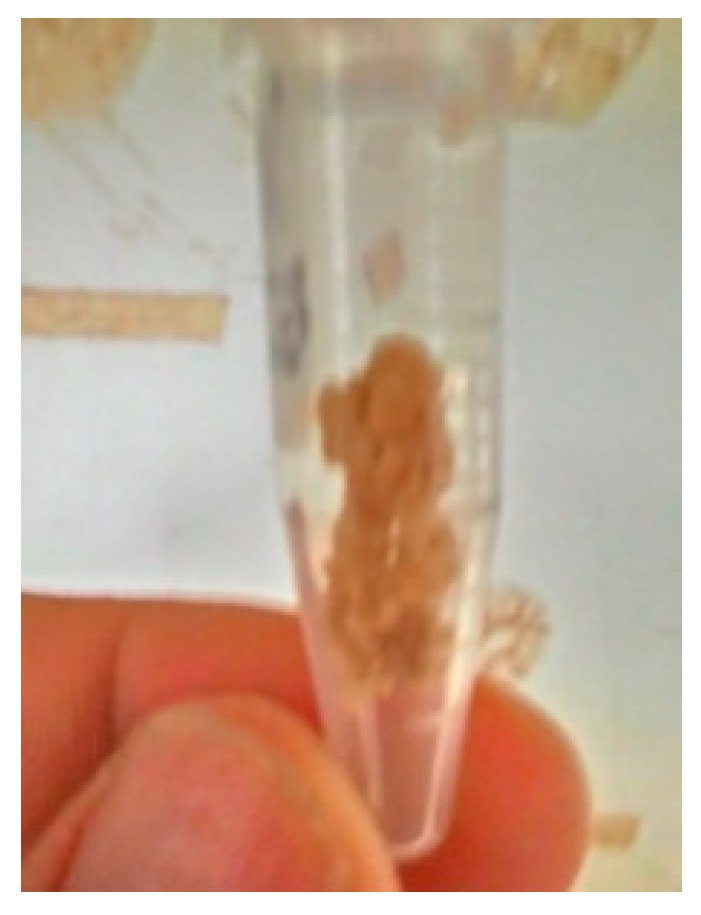
Confirmation of activity after sporulation of the strain (urease reaction).

**Figure 11 materials-12-04099-f011:**
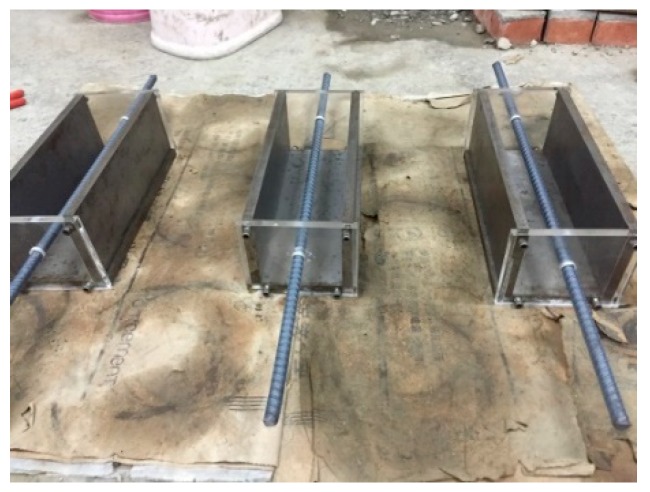
Mold of the beam specimen.

**Figure 12 materials-12-04099-f012:**
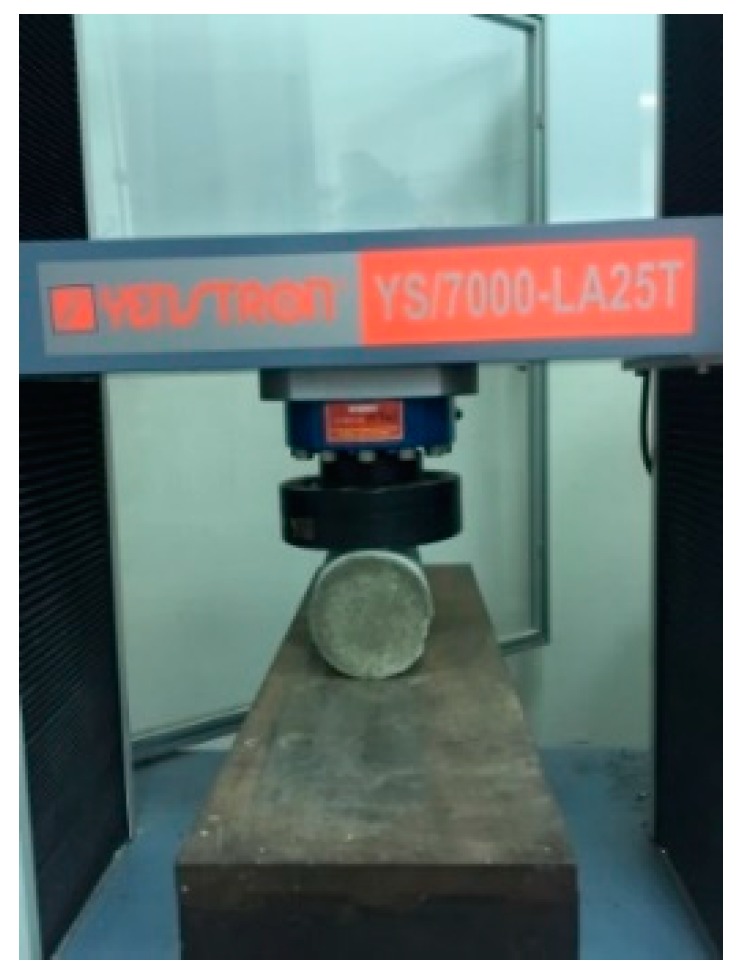
Pre-cracking of the concrete cylindrical specimen by the splitting method.

**Figure 13 materials-12-04099-f013:**
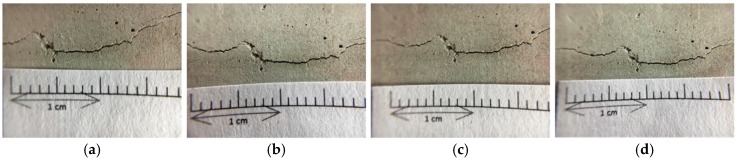
Images of the pre-cracked concrete cylindrical specimen (Control group): (**a**) Day 1; (**b**) Day 28; (**c**) Day 56; (**d**) Day 91.

**Figure 14 materials-12-04099-f014:**
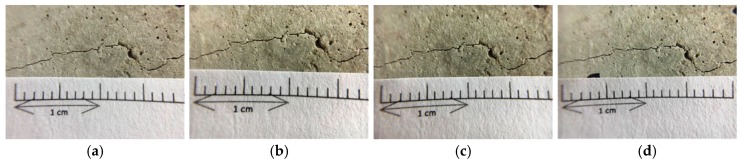
Images of the pre-cracked concrete cylindrical specimen (Experimental group I): (**a**) Day 1; (**b**) Day 28; (**c**) Day 56; (**d**) Day 91.

**Figure 15 materials-12-04099-f015:**
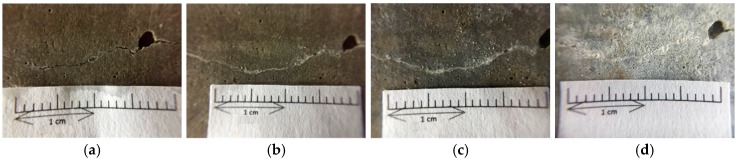
Images of the pre-cracked concrete cylindrical specimen (Experimental group II): (**a**) Day 1; (**b**) Day 28; (**c**) Day 56; (**d**) Day 91.

**Figure 16 materials-12-04099-f016:**
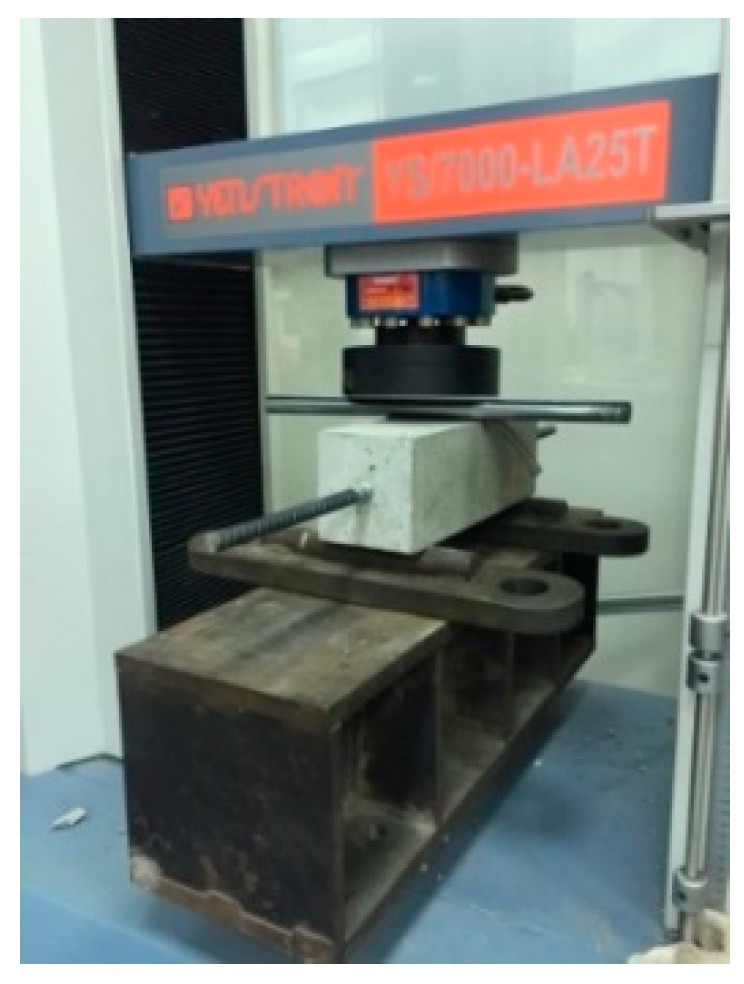
Pre-cracking of the beam specimen by the bending method.

**Figure 17 materials-12-04099-f017:**
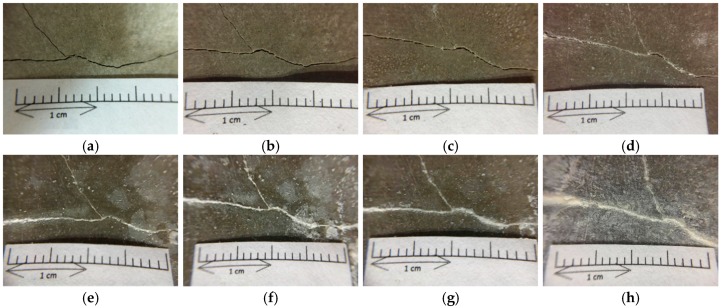
Images of the pre-cracked concrete beam specimen with small cracks (Experimental group II): (**a**) Day 1; (**b**) Day 3; (**c**) Day 7; (**d**) Day 14; (**e**) Day 21; (**f**) Day 28; (**g**) Day 56; (**h**) Day 91.

**Figure 18 materials-12-04099-f018:**
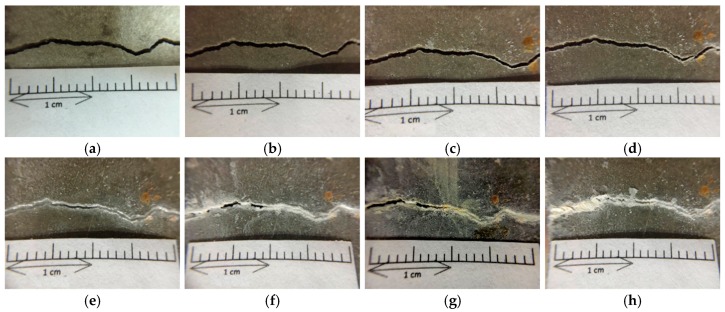
Images of the pre-cracked concrete beam specimen with large cracks (Experimental group II): (**a**) Day 1; (**b**) Day 3; (**c**) Day 7; (**d**) Day 14; (**e**) Day 21; (**f**) Day 28; (**g**) Day 56; (**h**) Day 91.

**Figure 19 materials-12-04099-f019:**
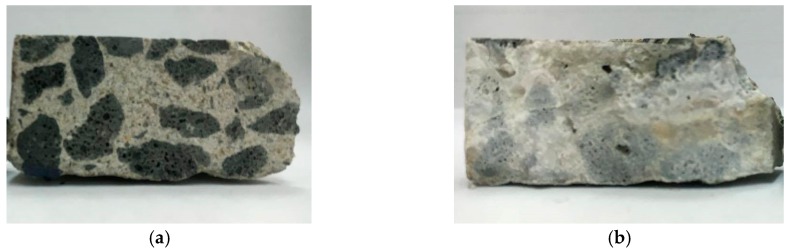
Comparison of cross-section observations between the control group and experimental group II: (**a**) control group; (**b**) experimental group II.

**Figure 20 materials-12-04099-f020:**
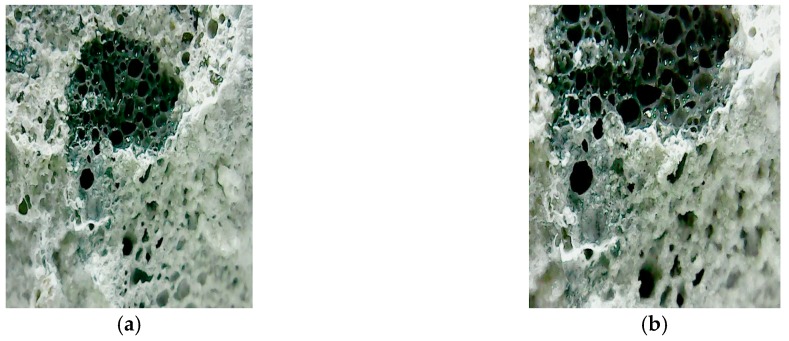
Sectional observation of experimental group II under a portable microscope: (**a**) 100 times magnification; (**b**) 200 times magnification.

**Figure 21 materials-12-04099-f021:**
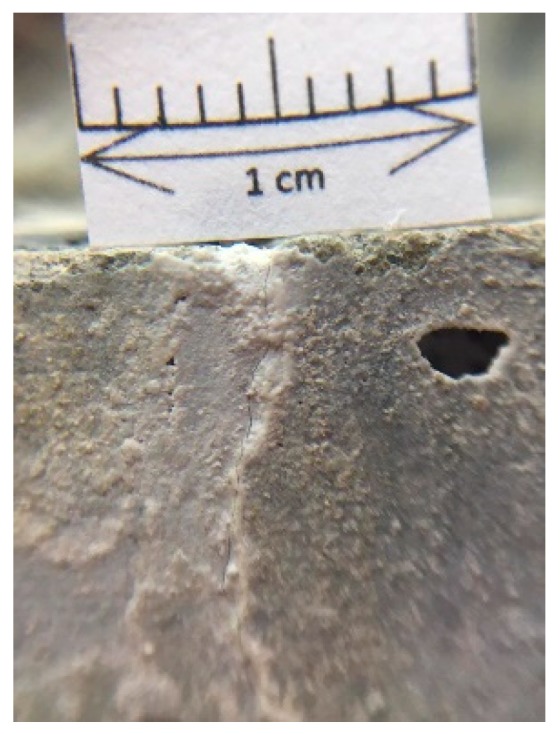
Surface crack of the concrete specimen.

**Figure 22 materials-12-04099-f022:**
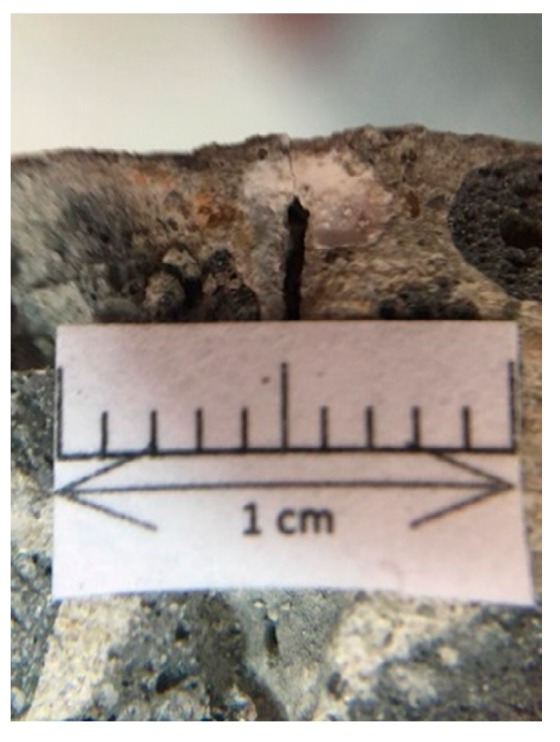
Crack repair of the concrete specimen.

**Figure 23 materials-12-04099-f023:**
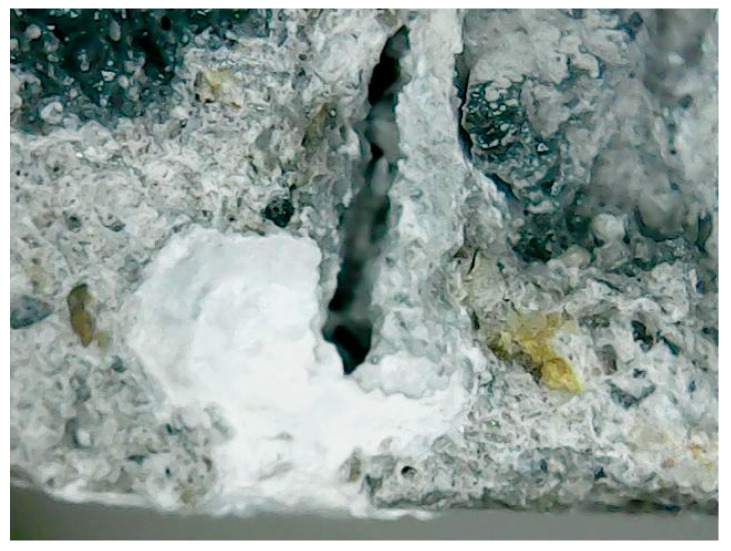
Thickness observation of the vertical section crack repair under a portable microscope (100 times magnification).

**Figure 24 materials-12-04099-f024:**
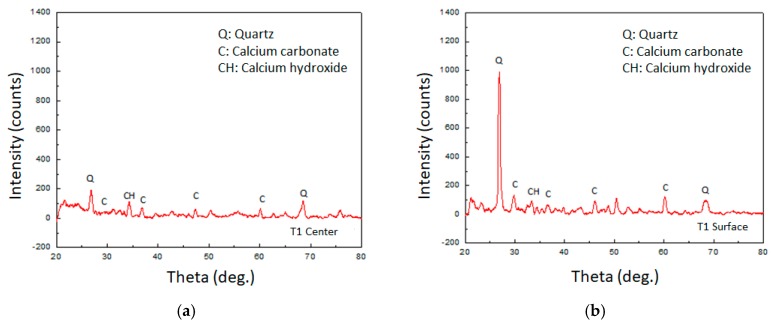
XRD test results of crack repair powder (experimental group I): (**a**) result at the sample center; (**b**) result at the sample surface.

**Figure 25 materials-12-04099-f025:**
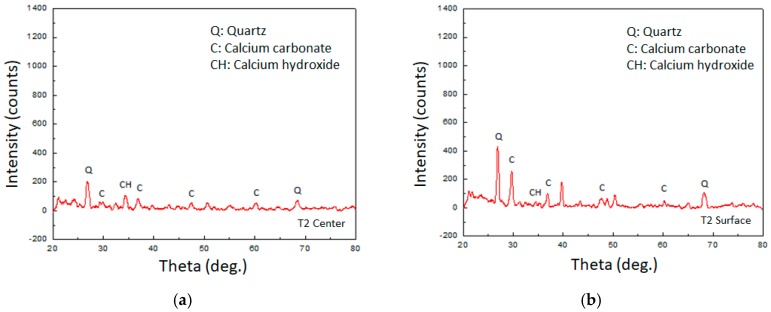
XRD test results of the crack repair powder (experimental group II): (**a**) result at the sample center; (**b**) result at the sample surface.

**Table 1 materials-12-04099-t001:** Basic properties of the lightweight aggregate.

Particle Density (g/cm^3^)	Water Absorption (%)		Loose Unit Weight (kg/m^3^)	Crushing Strength (MPa)	Porosity (%)
	1-hour	24-hour			
0.99	5.28	7.35	477.86	>3	24.97

**Table 2 materials-12-04099-t002:** Mix proportions of concrete.

Mix No.	W/C	Water (kg/m^3^)	Cement (kg/m^3^)	Lightweight Aggregate (kg/m^3^)	Sand (kg/m^3^)
LC	0.60	240	400	399	524

Notes: LC = lightweight aggregate concrete; W/C = water-to-cement ratio.

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
