# Peer review of "Self-Healing Concrete by Biological Substrate"

_materials, 2019, doi:10.3390/ma12244099_

Round 1

Reviewer 1 Report

This is a basic study of bacteria-aided self-healing of mortar (concrete) using Bacillus pasteurii. Self healing using Bacillus pasteurii has previously been conducted by several researchers, so I do not really see a significant originality in this work in its current presentation. If it is to be considered (I leave that to the editor to ultimately decide), the following items should be corrected/addressed:

The abstract has to be rewritten to improve its English and highlight more the originality. The abstract can do without the first two sentences. The lightweight aggregate is not adequately defined, what are the other physical properties, e.g. density, chemical composition. What is the effect of these aggregate on the availability of water for the cement. More details are required for the air curing e.g. what is the temperature, duration,.. It is not clear how many cylindrical and beam specimens were tested. This may have some impact on the reliability of the results. Quality of figures (24 and 25) should be improved. Although English can read ok there is a serious need for improvement in some places e.g. the sentence in line 119-122 is too long and the word “were” was used too many times.

Author Response

Point 1: This is a basic study of bacteria-aided self-healing of mortar (concrete) using Bacillus pasteurii. Self healing using Bacillus pasteurii has previously been conducted by several researchers, so I do not really see a significant originality in this work in its current presentation. If it is to be considered (I leave that to the editor to ultimately decide), the following items should be corrected/addressed:

The abstract has to be rewritten to improve its English and highlight more the originality. The abstract can do without the first two sentences. The lightweight aggregate is not adequately defined, what are the other physical properties, e.g. density, chemical composition. What is the effect of these aggregate on the availability of water for the cement. More details are required for the air curing e.g. what is the temperature, duration,.. It is not clear how many cylindrical and beam specimens were tested. This may have some impact on the reliability of the results. Quality of figures (24 and 25) should be improved. Although English can read ok there is a serious need for improvement in some places e.g. the sentence in line 119-122 is too long and the word “were” was used too many times.

Response:

The abstract has been rewritten to improve English writing and highlight more originality. The first two sentences in the original abstract have been removed. The basic properties of lightweight aggregates have been provided in the revised manuscript. Basically, these aggregates have larger pores and will absorb part of the mixing water. Therefore, in the mix design, the water-cement ratio has been increased. The revised manuscript has stated that the air curing of the specimens was performed in a curing room, the temperature of which was maintained at 23±5 °C, and the humidity was 100%. The revised manuscript has stated that a total of 48 cylindrical and 48 beam specimens was made. The quality of Figures 24-25 had been improved in the revised manuscript. The English writing of the revised manuscript has been reviewed by professionals.

Reviewer 2 Report

The manuscript by Chen et al., is well-written, the invetigation is appropriate, and the results are clearly presented. It looks promising also in view of future applications.

I recommend publication, only the following minor revisions are required:

Please insert, in the paragraph 2. Experimental Details, a sub-paragraph 2.4 describing the instrumental set-up you used for the performed experimental investigation (i.e. SEM, XRD), also removing some experimental details given in the Results and Discussion section. Lines 193-199: the english form should be improved, otherwise the sense of these sentences is not clear. 

Author Response

Point 1: The manuscript by Chen et al., is well-written, the investigation is appropriate, and the results are clearly presented. It looks promising also in view of future applications.

I recommend publication, only the following minor revisions are required:

Please insert, in the paragraph 2. Experimental Details, a sub-paragraph 2.4 describing the instrumental set-up you used for the performed experimental investigation (i.e. SEM, XRD), also removing some experimental details given in the Results and Discussion section. Lines 193-199: The English form should be improved, otherwise the sense of these sentences is not clear.

Response:

A sub-section (2.4. Instrument Setup and Test Procedures) has been added to Section 2 to illustrate the setup of the instrument. And some experimental details given in the Results and Discussion section has been moved to section 2.4. The English writing (including original Lines 193-199) of the revised manuscript has been reviewed by professionals.

Round 2

Reviewer 1 Report

the script has been improved however, the English still requires much more polishing.